# MULTI-TASK DENSE PREDICTIONS VIA UNLEASHING THE POWER OF DIFFUSION

**Yuqi Yang**[1,2,*], **Peng-Tao Jiang**[2,*,†], **Qibin Hou**[1,‡], **Hao Zhang**[2], **Jinwei Chen**[2], **Bo Li**[2]

[1]VCIP, School of Computer Science, Nankai University     [2]vivo Mobile Communication Co., Ltd
`yangyq2000@mail.nankai.edu.cn`

## ABSTRACT

Diffusion models have exhibited extraordinary performance in dense prediction tasks. However, there are few works exploring the diffusion pipeline for multi-task dense predictions. In this paper, we unlock the potential of diffusion models in solving multi-task dense predictions and propose a novel diffusion-based method, called TaskDiffusion, which leverages the conditional diffusion process in the decoder. Instead of denoising the noisy labels for different tasks separately, we propose a novel joint denoising diffusion process to capture the task relations during denoising. To be specific, our method first encodes the task-specific labels into a task-integration feature space to unify the encoding strategy. This allows us to get rid of the cumbersome task-specific encoding process. In addition, we also propose a cross-task diffusion decoder conditioned on task-specific multi-level features, which can model the interactions among different tasks and levels explicitly while preserving efficiency. Experiments show that our TaskDiffusion outperforms previous state-of-the-art methods for all dense prediction tasks on the widely-used PASCAL-Context and NYUD-v2 datasets. Our code is available at https://github.com/YuqiYang213/TaskDiffusion.

## 1 INTRODUCTION

Dense prediction tasks, such as semantic segmentation (Lu et al., 2017; Chen et al., 2017) and depth estimation (Bhat et al., 2021; Ranftl et al., 2021), play an important role in computer vision. These tasks perform image understanding from different aspects. For instance, the goal of the semantic segmentation task is to classify each pixel while the depth estimation task aims to predict the depth of every pixel in the real world. Because of the inherent differences in these tasks, most deep-learning-based methods concentrating on these tasks (Chen et al., 2017; Bhat et al., 2021) customize different network architectures for different tasks (Chen et al., 2023b).

Recently, generalist vision models (Chen et al., 2021; 2022a; Cheng et al., 2022; Chen et al., 2023b; Ji et al., 2023; Wang et al., 2023) emerged, which aim to design unified network architectures to address a set of vision tasks. Among these generalist methods, diffusion models (Sohl-Dickstein et al., 2015; Ho et al., 2020; Song et al., 2020) have shown great potential in addressing different dense prediction tasks because it can redefine different dense prediction tasks as a unified label-denoising task. Among these diffusion-based generalist models, DDP (Ji et al., 2023) decouples the encoder and decoder and performs the iterative denoising process only in the decoder phase to improve the inference efficiency. Compared to discriminative-based methods, generative-based methods can capture the underlying conditional distribution of the prediction explicitly (Le et al., 2024) and perform better in detail.

Although generalist models have shown advantages over customized models for different tasks, in real-world applications, such as autonomous driving and virtual reality, perception models are usually required to reason on a bunch of dense prediction tasks. Under these circumstances, previous generalist methods training one model per task need multiple forward inferences to generate the

---

*Work was done during interning at vivo. The first two authors share equal contributions.

†Project was led by Peng-Tao Jiang.

‡Corresponding author.

Figure 1: Comparisons between the task-specific diffusion process (left) and our proposed joint diffusion process (right). Our method encodes the labels for different tasks into one cross-task map and performs denoising with one cross-task diffusion decoder.

predictions for all tasks, making the inference stage less efficient. In addition, diffusion models are proven to be able to capture the underlying distribution of each single task, though, the potential to capture the cross-task relations is still waiting to be discovered. The cross-task relation is an essential point to improve the overall performance of different dense prediction tasks in a multi-task framework (Ye & Xu, 2022a; Brüggemann et al., 2021). These motivate us to investigate whether the diffusion-based generalist models can be extended to the field of multi-task dense predictions regarding their potential in processing various dense prediction tasks (Ji et al., 2023; Chen et al., 2023b; Saxena et al., 2023).

Directly applying diffusion models to the multi-task dense predictions has several obvious challenges. Firstly, denoising for multiple tasks separately hinders the diffusion model from digging the task relations. In addition, the target labels for different tasks are heterogeneous (e.g., discrete category labels for semantic segmentation and continuous labels for depth estimation). It needs cumbersome task-specific encoding designed for labels of different tasks (e.g., analog bits (Chen et al., 2022b) for the discrete labels). At last, diffusion models perform an iterative denoising process to generate the final predictions, which need several forward passes to output the final predictions for each task. When dealing with multiple tasks, performing multiple forward inferences for each task leads to a reduction in efficiency.

To address these challenges, we present a novel multi-task diffusion network, coined as TaskDiffusion. Our TaskDiffusion couples the denoising diffusion processes of different tasks into a joint denoising diffusion process in the decoder. This strategy utilizes the diffusion model to model the task relations explicitly in a coarse-to-fine process during denoising, which can bring overall performance improvements to all tasks. Specifically, our joint denoising diffusion process includes cross-task label encoding and cross-task diffusion decoder. For cross-task label encoding, we use embedding layers to encode different task labels and map the concatenation of these features into a cross-task map. This encoding strategy can convert heterogeneous labels from different tasks without cumbersome task-specific encoding methods. For cross-task diffusion decoder, we apply a cross-task diffusion decoder that is conditioned on the task-specific features extracted from different levels. Different from applying a task-specific denoising decoder for different tasks separately as done in previous works (Ji et al., 2023), our TaskDiffusion performs the diffusion process by explicitly modeling the task relations and the level relations, which is important in multi-task learning (Vandenhende et al., 2020). We present a brief comparison between our method and the task-specific diffusion process in Fig. 1.

To our knowledge, we are among the first to leverage diffusion models in fully-labeled multi-task dense prediction and our method can achieve notable improvements over previous methods. To verify the effectiveness of our method, we conduct comprehensive experiments on PASCAL-Context and NYUD-v2. Experiments show that our method outperforms the previous state-of-the-art methods on all the tasks. By shedding light on the effectiveness of diffusion models, we believe the potential of the diffusion-based method awaits to be further explored. We hope that our method will bring new insight to the community. In conclusion, the contributions of this paper are three-fold.

- We explore how to leverage diffusion models as an effective solver for multi-task dense prediction and propose a novel joint denoising diffusion process to capture the relationship among tasks.

- We propose a cross-task label encoding strategy to get rid of cumbersome task-specific encoding and a cross-task diffusion decoder for modeling the task relations and level relations explicitly.

- We conduct extensive experiments on the PASCAL-Context and NYUD-v2 benchmarks. Results show that TaskDiffusion outperforms previous state-of-the-art methods on all tasks.

## 2 RELATED WORK

**Multi-task learning for dense predictions.** The multi-task learning for dense predictions is a widely studied research field in computer vision. Most of the multi-task dense prediction methods can be divided into two categories (Vandenhende et al., 2021). The first category is the optimization-based methods (Kendall et al., 2018; Chen et al., 2018; Guo et al., 2018; Zhao et al., 2018) which focus on the balance of training signals from different tasks. The second category is the architecture-based methods (Ruder et al., 2019; Lu et al., 2017; Xu et al., 2018; Vandenhende et al., 2020; Ye & Xu, 2022a;b; 2023; Zhang et al., 2019; 2018; Zhou et al., 2020), which aims to design unified networks that can learn all the tasks jointly. Recently, more architecture-based methods (Chen et al., 2020; Bachmann et al., 2022; Tian et al., 2024; Zhang et al., 2023; Yang et al., 2024b; Wang et al., 2024) raised and achieve impressive performance. The architecture-based methods can be further divided into encoder-focused methods (Ruder et al., 2019; Lu et al., 2017; Bruggemann et al., 2020; Guo et al., 2020) and decoder-focused methods (Xu et al., 2018; Vandenhende et al., 2020; Brüggemann et al., 2021; Ye & Xu, 2022a;b; 2023; Zhang et al., 2019; 2018; Zhou et al., 2020; Li et al., 2023). Among them, PAPNet (Zhang et al., 2019) utilizes a per-task pixel affinity matrix and diffuses back into task features to spread the task relationship. MTINet (Vandenhende et al., 2020) puts emphasis on the task-correlation from different levels, and performs distillation from all levels. More recently, TaskExpert (Ye & Xu, 2023) introduces the mixture-of-expert technique into the decoder of multi-task dense predictions. All the aforementioned methods are based on discriminative-based methods, which directly learn the probability for each pixel. Unlike the above methods, our method is based on the generative diffusion models and can model the joint distribution of labels for different tasks into multi-task dense predictions.

**Diffusion models.** Diffusion and score-based generative models (Song et al., 2020; Sohl-Dickstein et al., 2015; Ho et al., 2020) have shown impressive performance and stability in image generation tasks against the previous methods (Goodfellow et al., 2014; Kingma & Welling, 2013). Following these image generation pipelines, many methods attempt to utilize its strong generative ability to construct high-dimensional distribution in other modalities, such as videos (Ho et al., 2022), audios (Kolesnikov et al., 2020) and text (Li et al., 2022). Varied with different tasks, a proper reformulation is always needed for diffusion-based methods (Chen et al., 2022b; Dieleman et al., 2022). For instance, Analog Bits (Chen et al., 2022b) uses the binary bits to convert the discrete task label into the continuous state, which can better suit the continuous diffusion process. More related to our work, MT-Diffusion (Chen et al., 2023a) takes auxiliary task input as conditions to guide the image generation process and predict the task from the inner features. It uses different kinds of encoders to encode different task labels into diffusion space. However, it can only handle at most two tasks simultaneously, while our method takes multi-task labels as a cross-task map and performs the joint denoising process. This enables our method to convert heterogeneous task labels into one continuous feature space and meanwhile can capture the relations among tasks.

**Diffusion models for dense predictions.** Witnessing the success of diffusion models in generative tasks, it is natural to extend this impressive ability into the field of perception. Regarding this, many methods (Chen et al., 2023b; Wang et al., 2023; Ji et al., 2023; Saxena et al., 2023; Lee et al., 2024) attempt to introduce diffusion models into various dense prediction tasks. Among them, DFormer (Wang et al., 2023) focuses on universal segmentation and views image segmentation as a generative task using noisy masks. DepthGen (Saxena et al., 2023) trains the diffusion models to estimate the depth with the incomplete depth label. DDP (Ji et al., 2023) designs a simple but effective diffusion framework for multiple dense visual prediction tasks. It decouples the encoder and decoder and only performs iterative diffusion in the decoding phase. Nevertheless, these methods only focus on performing one task per model. Our method extends diffusion models to multi-task dense predictions with high efficiency and good overall performance as well. In the topic of multi-task dense prediction, DiffusionMTL (Ye & Xu, 2024) leverages the diffusion process to rectify the noisy predictions in multi-task partially supervised learning. However, it focuses on denoising the noisy predictions resulting from the partially supervised label. In contrast, our method focuses on leveraging the diffusion model as an effective solver under fully supervised circumstances and capturing the task relation in the diffusion process.

## 3 METHOD

### 3.1 PRELIMINARIES

Before introducing our method, we give a brief introduction to diffusion models (Sohl-Dickstein et al., 2015; Ho et al., 2020; Song et al., 2020). The diffusion process includes a forward noising process and a reverse denoising process. The forward noising process gradually adds noise to the data sample to generate a noisy sample $z_t$, which can be formulated as:

$$z_t = \sqrt{\gamma(t)}z_0 + \sqrt{1-\gamma(t)}\epsilon, \tag{1}$$

where $\epsilon$ is the Gaussian noise and $t \in \{0, 1, ..., T\}$ indicates the temperate time. $\gamma(t)$ is a monotonically decreasing function to control the signal-to-noise ratio and the degree of corrosion. In the forward noising process, the original data $z_0$ is iteratively broken towards the pure Gaussian noise $z_T$. At the training stage, a denosing network $f_\theta(z, t)$ parameterized by $\theta$ is trained to predict $z_0$ from $z_t$ by minimizing an objective function, which is a $l_2$ loss most of the time. At the inference stage, the diffusion models perform the reverse denoising process. The neural network follows a Markovian way that recovers $z_0$ from the pure Gaussian noise $z_T$ iteratively. More specifically, the process of $z_T \rightarrow z_{T-\delta} \rightarrow ... \rightarrow z_0$ is achieved by applying the denoising network to $z_t$ and then using the predicted $\tilde{z}_0$ to make the transition to $z_{t-\delta}$ iteratively.

In perception tasks (Chen et al., 2023b; Ji et al., 2023), the diffusion models usually take the feature $\mathbf{x}$ as conditions to perform denoising. For example, in the semantic segmentation task, the diffusion models take both the noisy segmentation label $z_t$ and conditional feature $\mathbf{x}$ as input to perform the denoising process. The conditional diffusion process can be formulated as follows:

$$q_\theta(z_{0:T}|\mathbf{x}) = q(z_T) \prod_{t=0}^{T} q_\theta(z_{t-1}|z_t, \mathbf{x}), \tag{2}$$

where $q_\theta(\cdot)$ is implemented by the transition rule based on denoising network $f_\theta(z, t, \mathbf{x})$ that takes $\mathbf{x}$ as conditional input. Our method is based on the conditional diffusion models to perform the perception task. We propose cross-task label encoding that generates cross-task maps as input to the diffusion decoder. We utilize the multi-level features as the condition.

### 3.2 ARCHITECTURE

The overall framework of our TaskDiffusion is presented in Fig. 2. Our whole framework is composed of a pixel-level encoder, a cross-task label encoder, and a cross-task diffusion decoder.

**Pixel-level encoder.** The pixel-level encoder takes an image $\mathbf{I}$ as input and extracts multi-level features for each task $\{\mathcal{F}_i^s \in \mathbb{R}^{C \times H \times W} | i \in \{1, 2, ..., N\}, s \in \{1, 2, ..., S\}\}$, denoted as $\{\mathcal{F}_i^s\}$ in the following paper. $H$, $W$, and $C$ denote the height, width, and channels of one-level feature, respectively. $N$ indicates the total number of levels and $S$ indicates the total number of tasks. Specifically, following previous multi-task methods (Ye & Xu, 2022b;a), we first utilize a shared backbone ViT (Dosovitskiy et al., 2020) for all tasks to extract task-generic features. We select features from different layers of the backbone, denoted as $\{\mathbf{X}^l | l \in \{l_1, l_2, ..., l_N\}\}$. $l$ is the layer index, and $\mathbf{X}^l$ denotes the features from the $l$-th chosen backbone layer. The $N$-level features are fed into $S$ task-specific branches to generate the task-specific multi-level features $\{\mathcal{F}_i^s\}$. Each task-specific branch contains two stacked convolutional blocks, each of which consists of a $3 \times 3$ convolution followed by a batch normalization layer and a GeLU activation layer and a $1 \times 1$ convolution. To learn discriminative multi-level task-specific features across different tasks, we design task-specific auxiliary heads to generate intermediate predictions. These intermediate predictions are supervised by their corresponding task labels $\{\mathbf{K}_s\}$ and the task-specific branches will be updated by the gradient from their task. It is a common practice in multi-task dense prediction as presented in previous works (Ye & Xu, 2022a; Vandenhende et al., 2020; Brüggemann et al., 2021). The output of each task-specific branch acts as the conditions for the cross-task diffusion decoder.

**Cross-task label encoder.** In multi-task dense predictions, the model needs to learn multiple tasks with heterogeneous labels, such as discrete classification labels and continuous depth labels. However, as shown in previous studies (Chen et al., 2023b; 2022b), the diffusion process does not suit the discrete labels well. As a result, these labels need to be pre-processed (Chen et al., 2022b; Ji et al.,

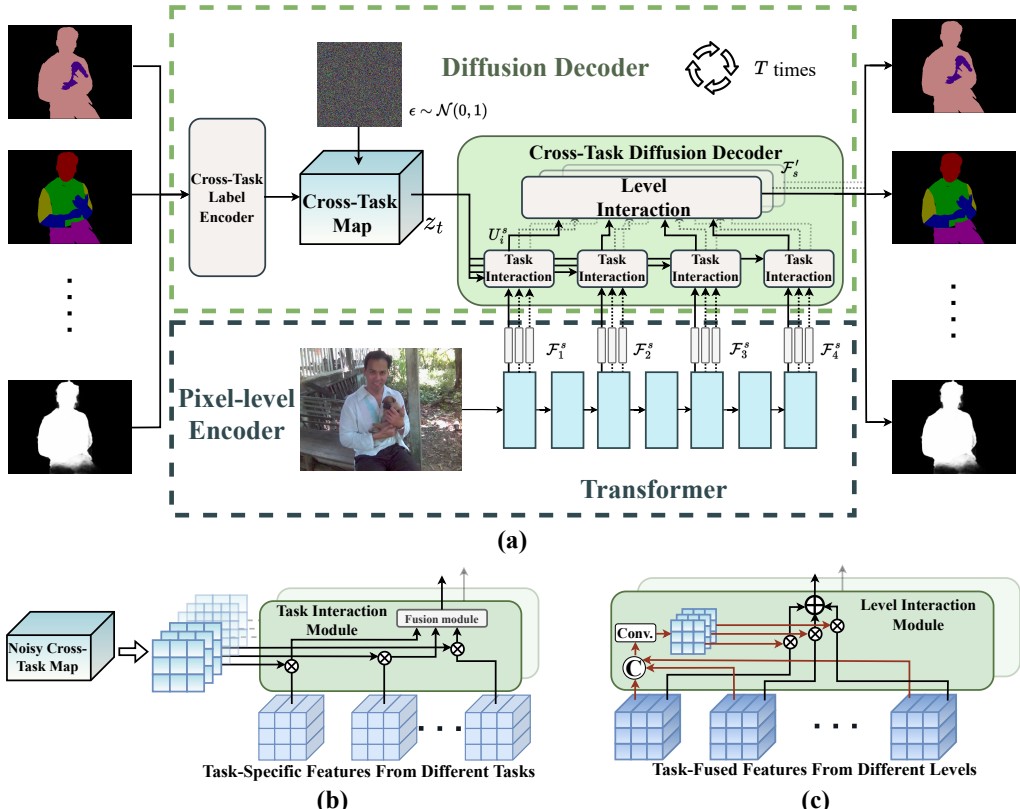

Figure 2: a) Overall framework of the proposed method. The cross-task diffusion decoder takes task-specific multi-level features as conditions and performs iterative denoising to the noisy cross-task map. The cross-task diffusion decoder is composed of task-interaction modules and level-interaction modules, which model the task relations and level relations explicitly and use them to combine the task-specific multi-level features from different tasks and different levels. The aggregated features are used to predict different tasks, and the predicted logits for each task are sent to the cross-task label encoder to generate the predicted cross-task map and perform iterative inference. b) Structure of the task interaction module. c) Structure of the level interaction module.

2023) separately. However, encoding the labels for different tasks into different feature spaces would prevent the model from capturing the relations among tasks, which are important in multi-task learning (Ye & Xu, 2022a; Brüggemann et al., 2021). In addition, designing task-specific encoding methods for different tasks is cumbersome and hard to generalize to different tasks. This motivates us to propose a cross-task label encoding mechanism to encode the heterogeneous labels for different tasks into one joint continuous feature space. This encoding mechanism also helps our diffusion decoder capture the task relations in one forward step.

Formally, we first encode different task labels $\{\mathbf{K}_s | s \in \{1, 2, ..., S\}\}$ to the feature space with a unified label encoder. For discrete labels, such as semantic segmentation labels, we first convert them to one-hot labels. For continuous labels, we input the label itself into the label encoder. The label encoder includes one $1 \times 1$ convolutional layer that maps the task labels to task-specific encoded label $\{\mathbf{K}'_s \in \mathbb{R}^{C \times H \times W} | s \in \{1, 2, ..., S\}\}$. To further capture the task relations, we concatenate the task-specific encoded labels of different tasks and use one $1 \times 1$ convolutional layer to map the concatenated task-specific encoded labels to the cross-task map $z$. This joint mapping process takes all the encoded maps into consideration and models task relations in a high-dimensional feature space. Following previous methods (Ji et al., 2023; Chen et al., 2023b), we normalize the cross-task map $z$ to $[-1, +1]$ and use one scaling factor $scale$ to control the signal-to-noise ratio. This helps us to enhance the difficulty of the denoising task and the diffusion decoder.

After encoding the labels for different tasks into the cross-task map $z$, we then add Gaussian noise to generate the corrupted mask $z_t$. As shown in Eqn. 1, $\gamma(t) \in [0, 1]$ controls the intensity of the

**Algorithm 1** TaskDiffusion training

```
def train(images, masks_gts):
  """
  images: [B, 3, H, W]
  masks_gts: {task:[B, *, H, W]}
  tasks: All the tasks that need to predict
  """
  # Encode task-specific multi-level
      features
  feats = pixel_encoder(images)

  # encoding the masks with different task-
      specific label encoder
  for task in tasks:
    m_enc[task] = ts_label_encoder[task](
        masks_gts[task])

  # encoding into one cross-task map
  m_enc_cross = label_encoder(cat(m_enc)) *
      scale

  # Corrupt the cross-task map according to
      the time
  t = randint(0, T) # timestep
  eps = normal(mean=0, std=1) # noise as
      standard gaussian
  m_crpt = sqrt(alpha_cumprod(t)) *
      m_enc_cross +
          sqrt(1 - alpha_cumprod(t)) * eps

  # Predict and compute loss
  m_preds = diff_decoder(m_crpt, feats, t)
  # using tasks-specific loss to train the
      model
  for task in tasks:
    loss[tasks] = prediction_loss[tasks](
        m_preds[task], masks_gts)

  return loss
```

**Algorithm 2** TaskDiffusion inference

```
def infer(images, steps):
  """
  images: [B, 3, H, W]
  steps: number of sample steps
  """
  # Extract task-specific multi-level
      features
  feats = pixel_encoder(images)

  # Generate noisy cross-task map
  m_t = normal(mean=0, std=1)

  for step in range(steps):
    # time interval generate
    t_now = 1 - step / steps
    t_next = max(1 - (1 + step + t_diff) /
        steps, 0) # asymmetirc time sampling

    # Predict m_preds from m_t
    m_preds = diff_decoder(m_t, feats, t_now
        )

    # encoding m_preds with task-specific
        encoder
    for task in tasks:
      m_enc[task] = ts_label_encoder[task](
          m_preds[task])

    # encoding into one cross-task map
    m_enc_cross = label_encoder(cat(m_enc))
        * scale

    # estimate map_t at t_next
    m_t = ddim(m_t, m_enc_cross, t_now,
        t_next)

  return m_preds
```

corruption noise which decreases with the increasing of time $t$. Following the previous method (Ji et al., 2023), we use a cosine schedule (Nichol & Dhariwal, 2021) for $\gamma(t)$ to control the noise.

**Cross-task diffusion decoder.** The decoder takes the noisy cross-task map $z_t \in \mathbb{R}^{C \times H \times W}$ as input and the task-specific multi-level features $\{\mathcal{F}_i^s\}$ as conditions. We also use the task-sharing feature $\mathbf{X}^l$ to help model the cross-task relation. The noisy label map $z_t$ is concatenated with $\mathbf{X}^l$ and sent to the cross-task diffusion decoder to perform task interaction and level interaction successively with $\{\mathcal{F}_i^s\}$.

Specifically, in the task-interaction phase, we perform task-interaction at each level. We utilize 2 convolutional blocks (Convolution-BatchNorm-GeLU-Convolution) and a convolutional layer to map the concatenated cross-task noisy map $z_t$ and $\mathbf{X}^l$ to each feature level and the channel number is transformed from $C$ to $S^2$. The task-relation maps are generated by applying a Sigmoid function to the output feature of the convolutional block at each level, denoted as $\{\mathbf{A}_i \in \mathbb{R}^{S \times S \times H \times W} | i \in \{1, 2, ..., N\}\}$. Then, the task-fused features are generated by multiplying $\{\mathbf{A}_i\}$ and the task-specific multi-level features $\{\mathcal{F}_i^s\}$, which is formulated as $\mathbf{U}_i^s = \sum_{p=1}^{S} \mathbf{A}_i^{s,p} \cdot \mathcal{F}_i^p$. The task-fused features are fed into the level-interaction phase.

In the level-interaction phase, the task-fused features $\{\mathbf{U}_i^s \in \mathbb{R}^{C \times H \times W} | i \in \{1, 2, ..., N\}\}$ of task $s$ from different levels are concatenated and delivered to another convolutional block that maps the channel number from $N \times C$ to $N$. The output is sent to a Sigmoid function to generate the task-specific level-fusing map $\mathbf{M}^s \in \mathbb{R}^{N \times H \times W}$. We use $\mathbf{M}^s$ to generate the final aggregated task-specific features by $\mathcal{F}_s' = \sum_{i=1}^{N} \mathbf{M}_i^s \cdot \mathbf{U}_i^s$, where $\mathcal{F}_s'$ indicates the aggregated task-specific features, $\mathbf{M}_i^s$ indicates the $i$-th element in the first dimension. The aggregated task-specific feature $\{\mathcal{F}_s' | s \in 1, 2, ..., S\}$ is sent to the task-specific prediction branch to generate the final predictions. Each task-specific branch contains three convolutional blocks. The predictions are then encoded as mentioned in Sec. 3.2 to generate the predicted $\tilde{z}_0$. Furthermore, we will discuss the advantages of our cross-task diffusion decoder in Sec. 3.3.

## 3.3 TRAINING AND INFERENCE

In the training phase, we first generate a cross-task map $z$ and add noises to get a noisy cross-task map $z_t$. This noisy map is inputted to the diffusion decoder. Then, we train the model to do the denoising process. The training algorithm is described in Algorithm 1. In the inference stage, the diffusion model takes an initial Gaussian noise as input and iteratively denoises this noisy map to get closer to the ground truth for different tasks. This procedure is summarized in Algorithm 2.

**Loss formulation.** In diffusion models for image generation tasks (Ho et al., 2020; Song et al., 2020), the $l_2$ loss is usually used. However, previous methods that adapt diffusion models for perception tasks (Chen et al., 2023b; Wang et al., 2023) show that the discriminative losses work better than the standard $l_2$ loss. Our method, different from the methods mentioned above, involves multiple tasks which are trained with different loss functions. As a result, we do not use $l_2$ loss to supervise the predicted $\tilde{z}_0$. Instead, we use the weighted task-specific loss (e.g. cross-entropy for semantic segmentation and $l_2$ loss for depth estimation) to supervise the predicted logits for different tasks. Specifically, our loss function can be formulated as:

$$\mathcal{L}_{all} = \sum_{s=1}^{S} w_s \mathcal{L}_s(k_s, K_s),\tag{3}$$

where $k_s$ is the predicted logits for task $s$, $w_s$ is the loss weight for task $s$ and $\mathcal{L}_s$ stands for the task-specific loss function for task $s$. We adopt the loss functions for different tasks following (Ye & Xu, 2022b) and will discuss them in detail in Sec. 4.1.

**Joint denoising.** In multi-task dense predictions, the decoder-focused methods (Ye & Xu, 2022b;a) always use one shared encoder and several task-specific decoders to generate the predictions for different tasks. Furthermore, the multi-level interaction is also important when constructing task relations (Vandenhende et al., 2020). As a result, it is natural to design several task-specific diffusion decoders that are conditioned on multi-level task-generic features. However, the design will result in degradation in both performance and efficiency. The reasons are two-fold. First, it is important to learn both the task-specific features, the task-generic features, and the task relations in the decoder (Ye & Xu, 2022b). The task-specific diffusion decoder makes it hard to model the task-generic features and the task relations. In contrast, our method performs a coarse-to-fine task-relation modeling in the cross-task diffusion decoder, as shown in Fig. 3. This helps our method to generate predictions with better accuracy in the iterative denoising process. Second, the diffusion decoder needs multiple forward passes to predict the final results. By joint denoising the different tasks, our method not only models the task relations in the diffusion decoder but also saves computational costs in the inference stage. We show the details of the time complexity in the appendix.

**Sampling strategy.** In our method, we choose DDIM (Song et al., 2020) as our map updating rule. After the $\tilde{z}_0$ is predicted for each time step, we use the reparameterization trick to generate the noisy cross-task map for the next step. Following (Ji et al., 2023; Chen et al., 2023b; 2022b), we use an asymmetric time intervals in inference. The time intervals can be controlled by t_diff in Algorithm 2, which is set to 1 empirically.

## 4 EXPERIMENTS

## 4.1 EXPERIMENTAL SETTINGS

**Datasets and evaluation metrics.** We conduct experiments on two public multi-task datasets, including PASCAL-Context (Chen et al., 2014) and NYUD-v2 (Silberman et al., 2012). The PASCAL-Context dataset contains 4,998 training images, 5,105 test images and annotations of five dense prediction tasks, including semantic segmentation, human parsing, saliency detection, surface normal prediction, and boundary detection. The labels for surface normal prediction and saliency detection are obtained from previous works Maninis et al. (2019). The NYUD-v2 dataset contains 795 training images, 654 test images, and annotations of four dense prediction tasks, including semantic segmentation, monocular depth estimation, surface normal prediction, and boundary detection. The input resolution for these two datasets are 512×512 and 448×576 respectively.

Following previous works (Ye & Xu, 2023; 2022b;a), we utilize the mean intersection-over-union (mIoU) to evaluate the semantic segmentation task and the human parsing task. The root mean square

Table 1: Quantitative comparison of different methods on PASCAL-Context dataset. † denotes the methods are reproduced based on the ViT-large backbone by us. * denotes the methods are reproduced based on the ViT-large backbone in (Ye & Xu, 2023). ** denotes the methods are reproduced based on fully labeled circumstances by us. Our method performs the best on all five tasks. ↑ denotes higher is better. ↓ denotes lower is better.

| Method | Semseg mIoU ↑ | Parsing mIoU ↑ | Saliency maxF ↑ | Normal mErr ↓ | Boundary odsF ↑ | $\Delta_m$ % ↑ | FLOPs (G) | #Params (M) |
|---|---|---|---|---|---|---|---|---|
| Single Task Learning | 81.62 | 72.21 | 84.34 | 13.59 | 76.79 | - | - | - |
| MTI-Net* (Vandenhende et al., 2020) | 78.31 | 67.40 | 84.75 | 14.67 | 73.00 | -4.62 | 774 | 851 |
| ATRC* (Brüggemann et al., 2021) | 77.11 | 66.84 | 81.20 | 14.23 | 72.10 | -5.50 | 871 | 340 |
| MQTransformer† Xu et al. (2023a) | 77.72 | 65.14 | 84.43 | 14.63 | 54.77 | -10.16 | 360 | 314 |
| DeMT† Xu et al. (2023b) | 78.96 | 67.39 | 84.26 | 14.53 | 55.29 | -8.99 | 372 | 308 |
| InvPT (Ye & Xu, 2022a) | 79.03 | 67.61 | 84.81 | 14.15 | 73.00 | -3.61 | 669 | 423 |
| TaskPrompter (Ye & Xu, 2022b) | 80.89 | 68.89 | 84.83 | 13.72 | 73.50 | -2.03 | 497 | 401 |
| TaskExpert (Ye & Xu, 2023) | 80.64 | 69.42 | 84.87 | 13.56 | 73.30 | -1.74 | 622 | 420 |
| DiffusionMTL** (Ye & Xu, 2024) | 80.46 | 69.13 | 84.85 | 14.02 | 70.96 | -3.16 | 732 | 381 |
| TSP-Transformer Wang et al. (2024) | 81.48 | 70.64 | 84.86 | 13.69 | 74.8 | -1.01 | 1991 | 422 |
| MLoRE (Yang et al., 2024b) | 81.41 | 70.52 | 84.90 | 13.51 | 75.42 | -0.62 | 571 | 407 |
| TaskDiffusion (**ours**) | 81.21 | 69.62 | 84.94 | 13.55 | 74.89 | -1.11 | 610 | 416 |
| TaskDiffusion (**ours**) /w MLoRE | **81.58** | **71.3** | **85.05** | **13.43** | **76.07** | **-0.04** | 738 | 472 |

Table 2: Quantitative comparison of different methods on the NYUD-v2 dataset. All the methods are based on ViT-large backbone.

| Method | Semseg mIoU ↑ | Depth RMSE ↓ | Normal mErr ↓ | Boundary odsF ↑ | $\Delta_m$ % ↑ |
|---|---|---|---|---|---|
| Single Task Learning | 56.77 | 0.5141 | 18.56 | 78.93 | - |
| InvPT (Ye & Xu, 2022a) | 53.56 | 0.5183 | 19.04 | 78.10 | -2.52 |
| TaskPrompter (Ye & Xu, 2022b) | 55.30 | 0.5152 | 18.47 | 78.20 | -0.81 |
| TaskExpert (Ye & Xu, 2023) | 55.35 | 0.5157 | 18.54 | 78.40 | -0.84 |
| TSP-Transformer Wang et al. (2024) | 55.39 | 0.4961 | 18.44 | 77.5 | -0.02 |
| MLoRE (Yang et al., 2024a) | 55.96 | 0.5076 | 18.33 | 78.43 | 0.11 |
| TaskDiffusion (**ours**) | 55.65 | 0.5020 | 18.43 | 78.64 | 0.18 |
| TaskDiffusion (**ours**) /w MLoRE | 56.66 | 0.5033 | 18.13 | 78.89 | 1.04 |

error (RMSE) metric is utilized to evaluate the monocular depth estimation task. The surface normal prediction task and boundary detection task are evaluated by the mean error (mErr) metric and the optimal-dataset-scale F-measure (odsF) metric, respectively. To evaluate the overall performance on all tasks, we utilize the MTL gain metric ($\Delta_m$) following (Maninis et al., 2019).

**Training and inference details.** Following previous works (Ye & Xu, 2022b;a; 2023), we utilize the ViT-large (Dosovitskiy et al., 2020) as our backbone and ViT-base for all ablation experiments. The batch size is set to 4 and all experiments are trained for 40000 iterations. The initial learning rate is set to 2e-5 for PASCAL-Context and 1e-5 for NYUD-v2. The weight decay is set as 1e-6 for both datasets. We use a polynomial learning rate scheduler following the previous method (Ye & Xu, 2022b). For the tasks (e.g. depth estimation and surface normal prediction) which have continuous labels, we use the $l_1$ loss. We use the cross-entropy loss for the other tasks with discrete labels (e.g., semantic segmentation, human parsing, saliency object detection, and boundary). To balance the training losses for different tasks, we follow the previous work (Ye & Xu, 2022b) to set the loss weights. In PASCAL-Context, we encode the labels for saliency object detection with a separate convolution and encode the other four tasks with the cross-task label encoder. The final cross-task map $z$ is the concatenation of these two encoded features. All the experiments are trained with 2 NVIDIA V100 GPUs for 40000 iterations.

## 4.2 Comparisons with state-of-the-art methods

We present the quantitative comparisons between the proposed method and previous state-of-the-art methods. Our method performs clearly better than most of the previous methods for all tasks on both PASCAL-Context and NYUDv2. The results can be found in Tab. 1 and Tab. 2. Since our proposed TaskDisffusion is a novel pipeline that leverages diffusion models to model the task relation explicitly, it can cooperate with the multi-task architecture and achieve higher performance. By

Table 3: Ablation study on the effectiveness of different components.

| Settings | Semseg mIoU ↑ | Parsing mIoU ↑ | Saliency maxF ↑ | Normal mErr ↓ | Boundary odsF ↑ | $\Delta_m$ % ↑ |
|---|---|---|---|---|---|---|
| Single Task Learning | 79.63 | 69.76 | 85.37 | 13.41 | 76.15 | - |
| Baseline | 77.34 | 66.17 | 85.18 | 13.78 | 72.40 | -2.80 |
| w/ Task-specific diffusion decoder | 77.41 | 67.15 | 85.29 | 13.83 | 74.40 | -2.02 |
| w/ Cross-task diffusion decoder | 78.51 | 67.32 | 85.26 | 13.47 | 74.60 | -1.11 |
| + Cross-task label encoding | 78.83 | 67.40 | 85.31 | 13.38 | 74.68 | -0.84 |

Table 4: Ablation on the impact of the task interaction and the level interaction in the cross-task diffusion decoder.

| Settings | Semseg mIoU ↑ | Parsing mIoU ↑ | Saliency maxF ↑ | Normal mErr ↓ | Boundary odsF ↑ | $\Delta_m$ % ↑ |
|---|---|---|---|---|---|---|
| Cross attention | 78.39 | 67.16 | 85.42 | 13.61 | 74.10 | -1.49 |
| Feature concatenation | 78.31 | 67.58 | 85.25 | 13.51 | 74.80 | -1.10 |
| Task interaction | 78.83 | 67.40 | 85.31 | 13.38 | 74.68 | -0.84 |
| w/o level interaction | 78.03 | 67.62 | 85.27 | 13.42 | 74.55 | -1.09 |
| w level interaction | 78.83 | 67.40 | 85.31 | 13.38 | 74.68 | -0.84 |

combining the previous method MLoRE Yang et al. (2024b) and our proposed TaskDiffusion, our method can outperform the previous method by a large margin. Particularly, the $\Delta_m$ outperforms MLoRE Yang et al. (2024b) and TSP-Transformer Wang et al. (2024) by +0.97% and +0.58% on the PASCAL-Context dataset. Furthermore, our method also achieves competitive efficiency compared to TaskExpert Ye & Xu (2023) (610GFLOPs vs 622 GFLOPs) although we run the inference step iteratively. In addition, we reproduce another diffusion-based multi-task dense prediction method DiffusionMTL (Ye & Xu, 2024) on fully-labeled circumstances. It can be seen that our method is clearly better than DiffusionMTL in aspects of both performance and computation cost with competitive parameter numbers. These facts demonstrate the effectiveness of the proposed joint denoising diffusion process. The visual results of our method and previous SoTA can be found in the appendix. Our method can generate better predictions, especially for semantic segmentation, human parsing, and boundary compared to the previous SoTA method, TaskPrompter.

## 4.3 ABLATION STUDY

The baseline in ablations uses the ViT-base as the backbone and extracts multi-level features directly from the 3-rd, 6-th, 9-th, and 12-th layers of ViT. The multi-level features from different levels are concatenated and then go through a task-specific branch, which contains one $1 \times 1$ convolutional layer and two convolutional blocks, to generate the fused feature $F_s^{fused}$. The fused feature is used to generate the final predictions. All ablation experiments are conducted on the PASCAL-Context dataset and run 3 steps of the denoising process to generate the final predictions in inference by default. The scaling factor $scale$ is set to 0.01 if not specifically specified. Some ablation on our method can be found in the appendix.

**Effectiveness of different components.** We ablate on the effectiveness of different components of our joint denoising diffusion process. The results are shown in Tab. 3. We first ablate the effectiveness of the diffusion decoder which is conditioned on the fused feature $F_s^{fused}$. We find that adding the task-specific diffusion decoder to the baseline will improve the MTL gain by a large margin. When replacing the plain task-specific diffusion decoder with the proposed cross-task diffusion decoder, the performance is further improved clearly. The performance gain can be attributed to the explicit modeling of the task relations and level relations in the cross-task diffusion decoder. The decrease in computation complexity can be attributed to the cross-task diffusion decoder, which will be discussed in the appendix. Finally, we add the cross-task label encoding and find that the performance on all tasks and the MTL gain increase, indicating the importance of capturing the task relations in the label encoding phase.

**Effectiveness of the task interaction and the level interaction.** We ablate on the effectiveness of the task interaction and the level interaction in the cross-task diffusion decoder. For the task interaction, we test three different ways of interaction. The first way is to apply cross-attention at different levels, where the cross-task noisy map is used as the query and the conditional features are as key and value. The second is directly concatenating the conditional features with the cross-task features. We generate the cross-task features from the cross-task noisy map $z_t$ by utilizing 2

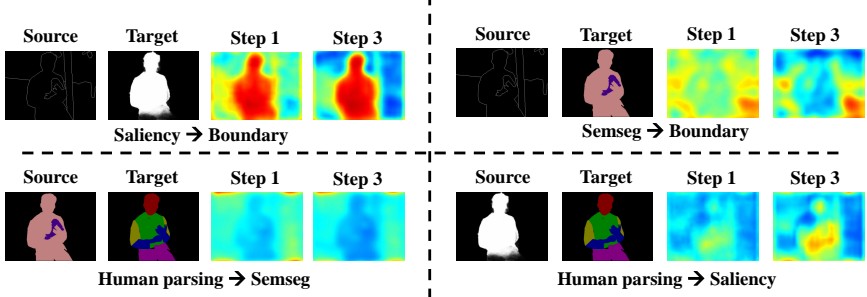

Figure 3: Visualizations of the task-relation mask $A_i$. In each pair of tasks, we show the task-relation mask of some pair of tasks in the first inference step and the third inference step. It can be observed that the task-relation mask in the third step focuses more on a specific area.

Table 5: Ablations on the scaling factor $scale$.

| $scale$ | Semseg mIoU ↑ | Parsing mIoU ↑ | Saliency maxF ↑ | Normal mErr ↓ | Boundary odsF ↑ | $\Delta_m$ % ↑ |
|---|---|---|---|---|---|---|
| 0.04 | 76.47 | 66.03 | 85.06 | 13.45 | 74.80 | -2.01 |
| 0.02 | 78.78 | 67.25 | 85.44 | 13.48 | 74.83 | -0.98 |
| 0.01 | 78.83 | 67.40 | 85.31 | 13.38 | 74.68 | -0.84 |
| 0.001 | 78.72 | 67.58 | 85.25 | 13.51 | 74.80 | -1.00 |

Table 6: Ablations on the inference steps.

| Steps | $\Delta_m$ % ↑ | FLOPs (G) |
|---|---|---|
| 1 | -1.47 | 508 |
| 3 | -0.84 | 667 |
| 5 | -0.83 | 827 |

convolutional blocks at each level. Both ways model the task relations implicitly by sharing the cross-task noise map. The third is our task interaction described in Sec. 3.2. The results are shown in Tab. 4. It can be seen that the task interaction performs better than both the cross-attention mechanism and simply concatenating the features, which implies the importance of explicitly modeling the task relations. In addition, we also conduct the ablation on the level interaction. It can be seen that after adding the level interaction, the performance of semantic segmentation is improved significantly from 78.03% to 78.83%. The $\Delta_m$ is also improved from -1.09% to -0.84%. These experimental results show the effectiveness of level interaction.

**Scaling factor.** We ablate on the scaling factor $scale$ mentioned in Sec. 3.2. The results are shown in Tab. 5. With the decrease of scaling factor, the overall performance of $\Delta_m$ increases accordingly and reaches the top when $scale$ is 0.01. We analyze that when the scaling factor becomes large, it is harder to train the model to denoise with heavily noisy samples, which will harm the diffusion model's denoising ability. When further decreasing the scaling factor to 0.001, the performance does not improve further.

**Inference steps.** We ablate on the number of the inference steps, and the results are shown in Tab. 6. With the increase of inference steps, the performance is improved by a large margin. Specifically, the performance of 3 steps for all tasks is higher than that of 1 step. When the inference step is further increased, there are no obvious improvements. Moreover, we can see that more inference steps also result in lower efficiency. In our method, we choose 3 steps that make a trade-off between performance and efficiency. To present an intuitive view of the effectiveness of iterative inference, we visualize the task-relation map $A_i$ for some task pairs from different levels, which is shown in Fig. 3. It can be seen that compared to the task-relation maps in the first step, the maps in the third step focus on a more specific area. These results show that our cross-task diffusion decoder performs a coarse-to-fine process with iterative inference. Further analysis can be found in the appendix.

## 5 CONCLUSIONS

In this paper, we propose a novel diffusion-based multi-task dense prediction method, coined as TaskDiffusion. To adapt the diffusion models to multi-task dense prediction, we propose a novel joint denoising diffusion process. Firstly, we encode the task-specific labels into the task-integration feature space. This unified encoding strategy eliminates cumbersome task-specific encoding and captures the task relation in label encoding. Furthermore, we propose the cross-task diffusion decoder conditioned on task-specific multi-level features. We model the interaction between different tasks and levels explicitly while preserving efficiency. Experiments demonstrate that our method clearly outperforms the previous methods on all tasks.

## ACKNOWLEDGMENT

This work was funded by NSFC (No. 62225604, 62176130), the Science and Technology Support Program of Tianjin, China (No. 23JCZDJC01050). The Supercomputing Center of Nankai University partially supported computation.

## REPRODUCIBILITY STATEMENT

We provide detailed experimental settings in Sec. 4.1. The training and test datasets are publicly available. In addition, our code will be public in the future.

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

# A APPENDIX

## A.1 TIME COMPLEXITY OF JOINT DENOISING

We compare the time complexity of the task-specific diffusion decoder and our cross-task diffusion decoder to show the efficiency of our proposed architecture. Taking multi-level task-specific decoder conditioned on multi-level backbone features $\{\mathbf{X}^l\}$ as an example, assume that each task contains two stacked convolutional blocks at different levels. The time complexity for one pass for these blocks in total is $O(2NS)$. Recall that $N$ denotes the number of levels and $S$ denotes the number of tasks. The features from each level are first concatenated with $z_t$ along the channel dimension and go through these two convolutional blocks. The output features from each level are then concatenated and the channel numbers are projected from $4C$ to $C$ by a $1 \times 1$ convolutional layer. The time complexity here can be ignorable for simplicity. After that, the projected feature goes through 3 stacked convolutional blocks to generate the final logits, where the time complexity is $O(3S)$. In the inference stage, each feature from $\{\mathbf{X}^l\}$ is concatenated with the same $z_t$ along the channel dimension, and the results from different feature levels are fed to the decoder to produce the denoised $z_{t-\delta}$. The task-specific diffusion decoder design results in $O(T(2NS + 3S))$ time complexity in total to generate $z_0$.

As for our joint denoising, we first generate the task-specific multi-level feature with the time complexity of $O(2NS)$. Then, for each pass, the cross-task map $z_t$ goes through the 2 convolutional blocks and a convolutional layer in each level to generate the task-relation maps with the time complexity of $O(2N)$. The generated task-relation maps are used to generate the task-specific multi-level features and the aggregated task-specific feature as discussed in Sec. 3.2. The time complexity for task interaction and level interaction can be ignorable for simplicity. The aggregated task-specific feature also goes through 3 stacked convolutional blocks to generate the final logits, where the time complexity is $O(3S)$. In total, our joint denoising achieves the complexity of $O(2NS + 2NT + 3ST)$, which saves the computation from multiple forward passes in the multi-level convolution module. More specifically, the FLOPs for our decoder is 568G with 3 forward passes compared to 779G for the task-specific diffusion decoder.

## A.2 VISUALIZATIONS OF OUR RESULTS

To give an intuitive comparison to other methods, we present the visual results in Fig. 4. It can be seen that our method has more accurate predictions on all tasks.

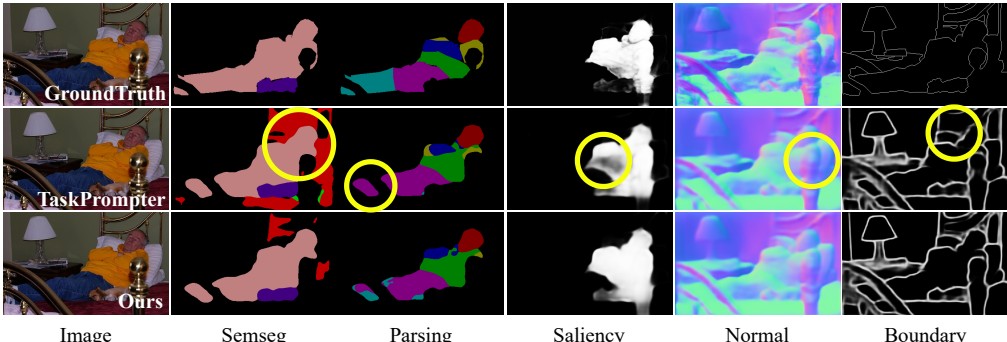

| Image | Semseg | Parsing | Saliency | Normal | Boundary |

Figure 4: Qualitative comparison with the previous SoTA (Ye & Xu, 2022b). Best viewed with zoom-in. It can be seen that our predictions achieve better results.

## A.3 VISUALIZATIONS OF THE CROSS-TASK MAP

We visualize the cross-task maps obtained by the label encoder to show what kinds of task information are extracted for different samples in Fig. 5. It can be seen that the cross-task maps obtained by the label encoder capture discriminate features for different tasks dynamically. For example, on the left of Fig. 5, the pixels with different human parsing labels are encoded to different features. It can be also seen that the pixels with different normal values (such as the pixels in the shoulder and the pixels in the body) are also encoded to different features. This supports that our cross-task map encoding

can encode the discriminative information from different tasks. In addition, on the right of Fig. 5, the differences in encoded features between pixels with different human parsing labels are less obvious. Instead, the encoded features for pixels with different semantic labels (human and dog) are more different from each other. This shows that our method can encode the cross-task map dynamically according to different samples.

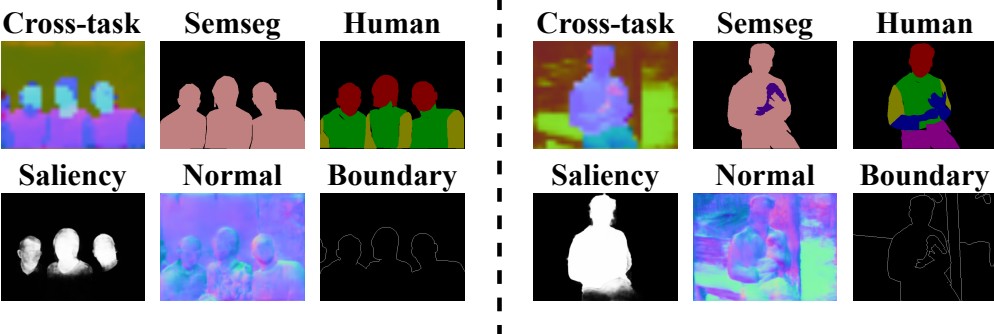

Figure 5: Visualizations of our cross-task map and labels for other tasks. The cross-task maps are visualized by PCA.

For a better understanding of the diffusion process, we also visualize the cross-task map at different steps. It can be seen from the Fig. 6 that the cross-task map in the first step is less accurate and more noisy compared to the cross-task map in the third step. This also supports that our diffusion process performs a coarse-to-fine process with interactive inference.

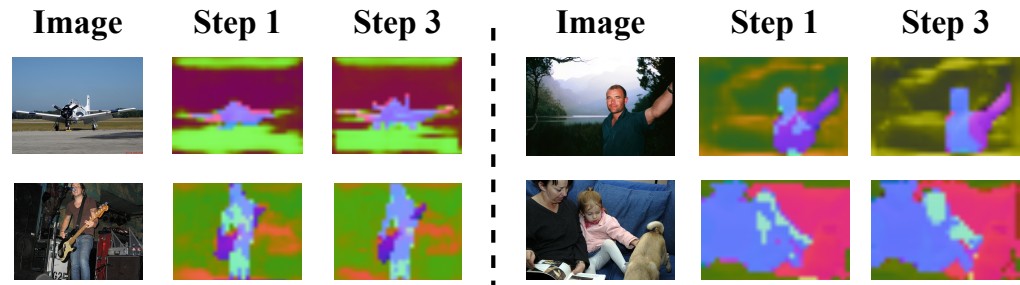

Figure 6: Visualizations of our cross-task maps in different steps. The cross-task maps are visualized by PCA.

### A.4 ANALYSIS ON THE VISUALIZED TASK-RELATION MASK

In Fig. 3, we visualize the $A_i^{s,p}$ mentioned in Sec. 3.2 for specific $s$, $p$ and $i$. We annotate task pairs on the bottom of each figure. For example, in the visualized $A_i^{s,p}$ in the left-top figure in Fig. 3, $p$ represents the task index for the saliency detection and $s$ represents the index for boundary detection. For a better understanding of the generated task-relation mask $A_i$, we will analyze all the visualized attention maps in Fig. 3 in the following.

First of all, to generate predictions with high quality, the neural network needs to extract various types of features from the original image. For instance, some features focus on boundaries, while others focus on the subject of the image. By synthesizing these features from different aspects together, the neural network can make an accurate prediction in different tasks. As a result, the cross-task attention map may not be directly correlated to the predictions, though, it will focus on the features that help improve the accuracy of the predictions. To utilize the diffusion model to model the task relation, we introduce the pixel-wise attention map $A_i^{s,p}$ generated in the diffusion process.

For example, in the left-top attention maps, saliency detection needs to detect the salient part of this image, which is the person. In addition, the feature of boundary detection needs to distinguish the

area inside the boundary from the outside. It will be useful for saliency detection to leverage the features inside the boundary of the person to distinguish the salient part of the human body from the background. As a result, the saliency detection task will pay more attention to the features of humans.

In addition, in the left-bottom attention maps, it can be seen that human parsing leverages the features from the background in semantic segmentation. Since the features of the person's area are required to be similar in different body parts, it is natural to ignore these features in human parsing, and the attention map also assigns low attention values to these areas. When it comes to the high attention value at the corner, we first need to delve into the details of inference. During inference, we pad the image to a resolution of 512×512 and the attention map assigns higher values to the padded area, which is also the background of this image. In the visualization, we cut the padded area for clarity, but some of the high-attention values remain in the cut image, which is the high-attention value at the corner.

In the right-top attention maps, they focus on the shadow on the ground and the bench on the left, where the boundaries are annotated around these areas. The semantic segmentation task can leverage these features inside the boundary extracted from the boundary detection branch to distinguish parts of the background from the area of the person and the dog. At last, the right-bottom attention maps focus on the hand of the person, which is also one of the segmentation targets of human parsing.

### A.5    AVOIDING NEGATIVE KNOWLEDGE TRANSFER

One challenge in multi-task learning is the task interference problem, which is also coined as negative knowledge transfer. The negative knowledge transfer among different tasks originates from receiving gradients to conflicting directions that cancel each other (Park et al., 2023). To mitigate this, we introduce the designed cross-task diffusion decoder, whose effectiveness can be captured in two ways. Firstly, we model the pixel-wise task relation explicitly in the decoder by learning the task-relation maps. For each task, our cross-task diffusion decoder will learn to activate task-specific features that are useful for the current task and suppress conflicted ones. In addition, our decoder can model different task relations in different diffusion steps, as shown in Fig. 3. This performs a coarse-to-fine task process with iterative inference, and it's also beneficial to generate fine-grained task relations. Secondly, we introduce task-specific modules, such as multi-level task-specific branches that are generated in our proposed model, which can also mitigate the negative transfer (Park et al., 2023). To make sure the multi-level task-specific branches generate task-specific features, we also add an auxiliary head to generate an intermediate prediction for different tasks and perform intermediate supervision. This auxiliary loss helps the multi-level task-specific branches to receive gradients for their tasks. It is a common practice in multi-task dense prediction as presented in previous works (Vandenhende et al., 2020; Ye & Xu, 2022a; 2024).

### A.6    EFFECTIVENESS ON SINGLE-TASK LEARNING

As a dense prediction method, our TaskDiffusion can also be adapted to single-task learning with some modification. When the task number is reduced to one, the cross-task relation map will be invalid since only one task exists. After removing this module, our method can perform single-task learning. We conduct experiments on our method in single-task learning and the results are listed in Tab. 7. It can be seen that the improvement compared to the single-task learning baseline is not as significant as that under multi-task learning circumstances. Since the proposed cross-task encoding is motivated by the limitation in performance in multi-task learning, our method will be more effective under multi-task learning circumstances by capturing the relation map across different tasks during the diffusion process.

Table 7: Ablation study on the effectiveness of TaskDiffusion on single-task learning.

| Settings | Semseg mIoU ↑ | Parsing mIoU ↑ | Saliency maxF ↑ | Normal mErr ↓ | Boundary odsF ↑ | $\Delta_m$ % ↑ |
|---|---|---|---|---|---|---|
| Baseline (STL) | 79.18 | 69.57 | 85.28 | 13.45 | 75.60 | - |
| TaskDiffusion (STL) | 79.54 | 70.71 | 85.35 | 13.37 | 77.19 | 0.97 |
| Baseline | 77.34 | 66.17 | 85.18 | 13.78 | 72.40 | -2.80 |
| TaskDiffusion | 78.83 | 67.40 | 85.31 | 13.38 | 74.68 | -0.84 |

### A.7    ROBUSTNESS AGAINST THE INITIAL SEED

Since our method is based on diffusion models, the inference of the proposed model will be influenced by the noise initilization. To evaluate the influence of initialization, we list three results generated with the same trained model under the different initial seeds. in Tab. 8. For efficiency, we use the loss value to evaluate the performance of boundary detection. It can be seen that the difference in performance is less than 0.01%, which is ignorable. This shows that our method is robust against different noise initialization.

Table 8: Ablation study on the Robustness against the initial seed.

| Method | Semseg ↑ | Parsing↑ | Sal.↑ | Nor. ↓ | Bound. loss ↓ |
|---|---|---|---|---|---|
| Seed 1 | 81.2058 | 69.6165 | 84.9360 | 13.5463 | 0.04273246 |
| Seed 2 | 81.2060 | 69.6164 | 84.9360 | 13.5463 | 0.04273244 |
| Seed 3 | 81.2058 | 69.6165 | 84.9361 | 13.5463 | 0.04273246 |

### A.8    LIMITATIONS

The limitations of our method are two-fold. First, our method can achieve better performance in a combination of 4 to 5 tasks. However, the performance for a combination of more tasks awaits to be studied. Second, our method focuses on tasks that share similarities, while the tasks with huge discrepancies (e.g.instance segmentation and classification) are less explored. We are looking forward to studying the diffusion models in more challenging multi-task settings.

