# OpenReview forum: "Multi-Task Dense Predictions via Unleashing the Power of Diffusion"
_ICLR.cc/2025/Conference — ICLR 2025 Poster_

### Official Review · Reviewer_QTvQ · 2024-10-29

**Soundness:** 3
**Presentation:** 3
**Contribution:** 3
**Rating:** 6
**Confidence:** 5

**Summary:**

The proposed TaskDiffusion encodes the task-specific labels into a task-integration feature space to unify the encoding strategy.In addition, the proposed cross-task diffusion decoder conditioned on task-specific multilevel features, which can model the interactions among different tasks and levels explicitly while preserving efficiency. The proposed TaskDiffusion’s experiment results also achieve comparable performance.

**Strengths:**

1 This paper is well-organized.

2 This paper leverages diffusion models as an effective solver for multi-task dense prediction.
The core idea of seeking a task-specific proxy to perform cross-task reasoning via diffusion model is intuitive.

3 It outperforms the state-of-the-art in most cases.

**Weaknesses:**

1 The value of \Delta_m (MTL gain metric) requires the value of  the single task (ST) model. It is suggested that the authors release the value of ST in Table 3. Following the calculation criteria of MTL on \Delta_m metric, showing Baseline ST and MT values will have a beneficial impact on the development of MTL. The reviewer saw Table 7, but it appeared in the appendix rather than in the main paper.

2 From the results in Table 3, the core diffusion decoder designed in this paper does not seem to bring much growth to the model. Table 3 also reveals that a good encoder leads to greater performance gains. The relative performance improvement brought by the proposed Cross-Task Diffusion Decoder is not even comparable to ATRC and InvPT.

3 Task-fused features U are generated by A$\cdot$F in line 302. Why the features after A$\cdot$F are fused features? The same question arises on line 308. Why can this matrix multiplication represent feature fusion and what is its mathematical interpretation?
Both task-fused features and task-fusing features mean the same map U.Moreover, the same problem has $p$ in 302 line and $p$ in Eq.2.These inconsistency are confusing to the reader.

4 To ensure comprehensive evaluation, it is imperative that the study provides a comparative analysis of the proposed technique against the most recent methods ("DeMT: Deformable Mixer Transformer for Multi-Task Learning of Dense Prediction" AAAI 2023; "Multi-Task Learning with Multi-Query Transformer for Dense Prediction" TCSVT 2024; "Multi-Task Dense Prediction via Mixture of Low-Rank Experts" CVPR 2024). In addition, the discussion of the decoder-based MTL approach in the related work section lacks an introduction to recent works, such as DeMT AAAI’23, 3DAwareMTL ICLR’24, MQTransformer TCSVT’24 and MLoRE CVPR’24.

5 Figure 4 does not seem to demonstrate the visualization advantage of the proposed TaskDdiffusion model in the appendix, compared to TaskPrompter.

**Questions:**

1 The authors state: “To our knowledge, we are among the first to leverage diffusion models in …”. The DiffusionMTL (CVPR’24) model already took advantage of the diffusion model.

2 What is Cross-Task Label Encoder based on?  Some tasks are labeled with images or data in matlab format. How do authors encode these different data formats?

---

> ### Author Response · Authors · 2024-11-21
>
> **To reviewer QTvQ**:
> Thank you for your valuable comment. We will address your concern in the following point-by-point.
>
> **Q1**: The value of Delta\_m (MTL gain metric) requires the value of the single task (ST) model. It is suggested that the authors release the value of ST in Table 3. Following the calculation criteria of MTL on Delta\_m metric, showing Baseline ST and MT values will have a beneficial impact on the development of MTL. The reviewer saw Table 7, but it appeared in the appendix rather than in the main paper.
>
> **A1**: Thank you for your suggestion. We agree with the importance of the performance of the ST model. We have added them to Tab.3 in the revisions.
>
> **Q2**: From the results in Table 3, the core diffusion decoder designed in this paper does not seem to bring much growth to the model. Table 3 also reveals that a good encoder leads to greater performance gains. The relative performance improvement brought by the proposed Cross-Task Diffusion Decoder is not even comparable to ATRC and InvPT.
>
> **A2**: It can be seen from Tab. 3 that the proposed cross-task diffusion decoder brings an improvement of 1.69% in $\Delta_m$ compared to the baseline, and 0.91% compared to the task-specific diffusion decoder. Following previous works [1], a performance gain in $\Delta_m$ for about 1.5%(e.g. In Tab. 1 the performance gain after +SPrompt is 1.52%) can be considered effective. In addition, our method focuses on proposing a better decoder that can improve the performance with tolerable efficiency loss, though, changing the encoder to a heavier backbone will bring more computation cost and parameter number. It is hard for a decoder to bring as much performance gain as a heavier encoder does. At last, the InvPT [1] is the first multi-task dense prediction method that leverages the ViT-large as the backbone. The performance of ATRC with ViT-large listed in Tab. 1 is reproduced in Taskprompter [2]. As a result, it is natural for InvPT, a method designed for the ViT-based backbone, to have a significant performance gain compared to ATRC, whichis designed for the CNN-based backbone.
>
> **Q3**: Task-fused features U are generated by A\_F in line 302. Why the features after A\_F are fused features? The same question arises on line 308. Why can this matrix multiplication represent feature fusion and what is its mathematical interpretation? Both task-fused features and task-fusing features mean the same map U.Moreover, the same problem has $p$ in 302 line and $p$ in Eq.2.These inconsistency are confusing to the reader.
>
> **A3**: In line 302, the $A_F$ represents the task-relation maps. More specifically, $A^{s, p}$ represents the weight for task $s$ to gain from the task-specific feature from task $p$. In other words, the task-fused feature $U^s$ for task $s$ is the weighted sum of task-specific features from different tasks, like the attention mechanism. In addition, the inconsistency of task-fused/task-fusing feature and annotation $p$ has been corrected in the revision.

---

> ### Author Response · Authors · 2024-11-21
>
> **Q4**: To ensure comprehensive evaluation, it is imperative that the study provides a comparative analysis of the proposed technique against the most recent methods (“DeMT: Deformable Mixer Transformer for Multi-Task Learning of Dense Prediction” AAAI 2023; “Multi-Task Learning with Multi-Query Transformer for Dense Prediction” TCSVT 2024; “Multi-Task Dense Prediction via Mixture of Low-Rank Experts” CVPR 2024). In addition, the discussion of the decoder-based MTL approach in the related work section lacks an introduction to recent works, such as DeMT AAAI’23, 3DAwareMTL ICLR’24, MQTransformer TCSVT’24 and MLoRE CVPR’24.
>
> **A4**: Thank you for your advice. Since some of the methods listed do not have performance based on ViT-large, we reproduce them based on ViT-large, and the performances are listed below. In addition, our method serves as a novel pipeline that leverages diffusion models to model the task relation explicitly. It can cooperate with other network architectures to achieve better performance. As a result, we combine the module proposed by MLoRE [1] and our method and it can surpass the performance of the previous SoTA on all tasks, especially in human parsing, normal prediction, and boundary detection. These results have been updated to Tab. 1 in the revision.
>
>  | Method | Semseg $\uparrow$ | Parsing$\uparrow$ | Sal.$\uparrow$ | Nor. $\downarrow$ | Bound.$\uparrow$ |
>  | ---------------- | ----------- | ------------ | ---------- | ------------ | ------------ |
>  | DeMT w/ ViT-Large | 78.96 | 67.39 | 84.27 | 14.53 | 55.29 |
>  | MQ-Transformer w/ViT-Large | 77.72 | 65.14 | 84.44 | 14.63 | 54.77 |
>  | MLoRE | 81.41 | 70.52 | 84.90 | 13.51 | 75.42 |
>  | ours w/ MLoRE | 81.58 | 71.30 | 85.05 | 13.43 | 76.07 |
>
> **Q5**: Figure 4 does not seem to demonstrate the visualization advantage of the proposed TaskDiffusion model in the appendix, compared to TaskPrompter.
>
> **A5**: The differences between our method and TaskPrompter are circled in the figure. For example, the TaskPrompter mistook the background behind the lying person for the chair, while our method can predict the background accurately. It can be seen that the results predicted by our method are closer to the ground truth than TaskPrompter on all the tasks.
>
> **Q6**: The authors state: “To our knowledge, we are among the first to leverage diffusion models in …”. The DiffusionMTL (CVPR’24) model already took advantage of the diffusion model.
>
> **A6**: Thank you for your comment. Our method is among the first to leverage diffusion models in fully-labeled multi-task dense prediction, while DiffusionMTL focuses on partially-labeled multi-task dense prediction. We have modified our statement into “We are among the first to leverage diffusion models in fully-labeled multi-task dense prediction ...” in the revision.
>
> **Q7**: What is Cross-Task Label Encoder based on? Some tasks are labeled with images or data in matlab format. How do authors encode these different data formats?
>
> **A7**: Our codes are based on Python. Following the code provided by previous works [1，2], we convert all the labels into 4D pytorch tensor for the following process.
>
> [1] Hanrong Ye and Dan Xu. Inverted pyramid multi-task transformer for dense scene understanding. In ECCV, pages 514–530. Springer, 2022a.
>
> [2] Hanrong Ye and Dan Xu. Taskprompter: Spatial-channel multi-task prompting for dense scene understanding. In ICLR, 2022b.
>
> [3] Yuqi Yang, Peng-Tao Jiang, Qibin Hou, Hao Zhang, Jinwei Chen, and Bo Li. Multi-task dense prediction via mixture of low-rank experts

---

> ### Comment · Reviewer_QTvQ · 2024-11-27
>
> Thanks the authors for the response. Most of the my concerns have been addressed.
>
> Please ensure that the additional content from your rebuttal are incorporated into the main text of your paper. This will enhance the clarity and completeness of your work, ensuring it effectively communicates your contributions to the community. Adhering to this commitment will significantly strengthen your paper.
>
> I decide to keep my original rating.

---

> > ### Author Response · Authors · 2024-11-27
> >
> > Thank you for your efforts that have helped us improve the quality of the paper greatly. If there are further questions, we will be very happy to discuss them.

---

### Official Review · Reviewer_T3fK · 2024-11-01

**Soundness:** 3
**Presentation:** 2
**Contribution:** 2
**Rating:** 6
**Confidence:** 5

**Summary:**

This paper presents a diffusion-based method for multi-task dense prediction, leveraging the conditional diffusion process. The method effectively models task relationships during the denoising process through the cross-task diffusion decoder and cross-task label encoder. The experiments demonstrate its effectiveness on two benchmark datasets.

**Strengths:**

1. The proposed framework facilitates joint diffusion denoising of multiple tasks, allowing the diffusion process to effectively capture cross-task relationships, which is beneficial for multi-task learning.
2. The method demonstrate strong performance on two benchmark datasets, PASCAL-Context and NYUD-v2.

**Weaknesses:**

1. The proposed modules have limited novelty. The cross-task diffusion decoder builds upon existing works, such as DDP [1], while incorporating interaction techniques from multi-task learning. The cross-task label encoder functions as an inverse process of task-specific heads in standard multi-task models. It would be better if the paper could demonstrate more insights in the utility of diffusion for multi-task dense prediction.
2. The authors are recommended to compare their method with more state-of-the-art methods, especially those published in 2024, such as [2, 3].
3. There is room for improvement in presentation, including language and figures. For example, the text in Figure 1 right is too small to be easily readable.

[1] Yuanfeng Ji, Zhe Chen, Enze Xie, Lanqing Hong, Xihui Liu, Zhaoqiang Liu, Tong Lu, Zhenguo Li, and Ping Luo. Ddp: Diffusion model for dense visual prediction. arXiv preprint arXiv:2303.17559, 2023.
[2] Yuqi Yang, Peng-Tao Jiang, Qibin Hou, Hao Zhang, Jinwei Chen, and Bo Li. Multi-task dense prediction via mixture of low-rank experts. In CVPR, 2024.
[3] Shuo Wang, Jing Li, Zibo Zhao, Dongze Lian, Binbin Huang, Xiaomei Wang, Zhengxin Li, and Shenghua Gao. TSP-Transformer: Task-specific prompts boosted transformer for holistic scene understanding. In WACV, 2024.

**Questions:**

1. In Section 3.2, task-specific auxiliary heads are used to generate intermediate predictions that are supervised by ground truth labels. It is interesting to know the contribution of these auxiliary heads to the final results, as they have different training procedure with the diffusion process.
2. In Algorithm 2, what is the relationship between masks_pred and m_preds?
3. In Table 1, the FLOPs of the proposed method with a ViT-large backbone is reported as 610G. However, in Table 6, the ablation study indicates that the model with 3 steps utilizes 667G FLOPs. Why does the model with the ViT-base backbone has a higher computational cost than the ViT-large backbone?

---

> ### Author Response · Authors · 2024-11-21
>
> **To reviewer T3fK:**
>
> Thank you for your valuable comment. We will address your concerns in the following point-by-point.
>
> **Q1**: The proposed modules have limited novelty. The cross-task diffusion decoder builds upon existing works, such as DDP, while incorporating interaction techniques from multi-task learning. The cross-task label encoder functions as an inverse process of task-specific heads in standard multi-task models. It would be better if the paper could demonstrate more insights in the utility of diffusion for multi-task dense prediction.
>
> **A1**: Thank you for your comment. The novelty of our method can be summarized in two points. Firstly, the diffusion models are proven to have outstanding ability in capturing the underlying distribution of each single task in DDP. However, the ability to capture the cross-task relations is overlooked by the existing method. In addition, the cross-task relation is the key point to improve the overall performance of different dense prediction tasks in multi-task learning. To achieve our goal, we designed the cross-task diffusion decoder that leverages the diffusion process to model the task relation explicitly in a coarse-to-fine manner. In other words, our method is inspired by the impressive performance of the diffusion process and leverages the diffusion process in the key problem of multi-task learning.
>
> Secondly, the cross-task label encoder is proposed to unify the encoding of discrete labels and continuous labels. The inverse process of task-specific heads and our cross-task label encoder is different in the target feature space. The inverse process of task-specific heads maps the different labels to their corresponding feature spaces, which are also task-specific. On the contrary, our cross-task label encoder maps the continuous or discrete spatial map to a feature space shared by all the tasks. This encoding mechanism can get rid of cumbersome task-specific encoding and encode the discriminative information from Wdifferent tasks to gain better performance.
>
> **Q2**: The authors are recommended to compare their method with more state-of-the-art methods, especially those published in 2024, such as MLoRE and TSP-Transformer.
>
> **A2**: Thank you for your suggestion. Our method proposes a novel diffusion-based multi-task learning pipeline that can be used to improve the task-relation generation in other multi-task learning networks. As a result, we combine the MLoRE module proposed in previous work [1] with our proposed pipeline and achieve a new SoTA in the PASCAL-Context dataset. The performance of it and previous SoTA are listed below. It can be seen from the result that our method can surpass the performance of the previous SoTA on all tasks, especially in human parsing, normal prediction, and boundary detection. These results have been updated to Tab.1 in the revision.
>
>
>  | Method | Semseg $\uparrow$ | Parsing$\uparrow$ | Sal.$\uparrow$ | Nor. $\downarrow$ | Bound.$\uparrow$ |
>  | ----------- | -------- | ----------- | -------- | --------- | ----------- |
>  | MLoRE | 81.41 | 70.52 | 84.90 | 13.51 | 75.42 |
>  | TSP-Transformer | 81.48 | 70.64 | 84.86 | 13.69 | 74.80 |
>  | ours w/ MLoRE | 81.58 | 71.30  | 85.05 | 13.43 | 76.07 |
>
> **Q3**: There is room for improvement in presentation, including language and figures. For example, the text in Figure 1 right is too small to be easily readable.
>
> **A3**: Thank you for your advice. We will polish the language and figures carefully in the camera-ready version. In addition, the text in Fig. 1 has been bigger in the revision.

---

> ### Author Response · Authors · 2024-11-21
>
> **Q4**: In Section 3.2, task-specific auxiliary heads are used to generate intermediate predictions that are supervised by ground truth labels. It is interesting to know the contribution of these auxiliary heads to the final results, as they have different training procedure with the diffusion process.
>
> **A4**: The task-specific auxiliary heads can generate intermediate predictions from task-specific branches. Without the auxiliary heads, there will be no guarantee that these branches are capturing task-specific features. As a result, it will be hard for the cross-task diffusion decoder to generate feasible task-relation which results in performance degradation. The performance of our method without the auxiliary heads is listed below. It can be seen that the auxiliary head brings a performance gain of 0.29% in $\Delta_m$, which supports our argument.
>
>  | Method | Semseg$\uparrow$ | Parsing$\uparrow$ | Sal.$\uparrow$ | Nor. $\downarrow$ | Bound.$\uparrow$ | $\Delta_m$ $\uparrow$ |
>  | ---------- | ----------- | ----------- | ---------- | ------------ | --------- | ---------- |
>  | ours w/o aux | 78.55 | 67.03 | 85.36 | 13.44 | 74.54 | -1.13 |
>  | ours w/ aux | 78.83 | 67.40 | 85.31 | 13.38 | 74.68 | -0.84 |
>
> **Q5**: In Algorithm 2, what is the relationship between masks\_pred and m\_preds?
>
> **A5**: Thanks for your comment. The “masks\_pred” in Algorithm 2 should be “m\_preds”. We have corrected this typo in the revision.
>
> **Q6**: In Table 1, the FLOPs of the proposed method with a ViT-large backbone is reported as 610G. However, in Table 6, the ablation study indicates that the model with 3 steps utilizes 667G FLOPs. Why does the model with the ViT-base backbone has a higher computational cost than the ViT-large backbone?
>
> **A6**: In the ViT-base backbone, we follow the previous method [2,3] and use more channels in the decoder. In multi-task dense prediction, since the decoder is branched for different tasks, the computation costs for the decoder can outweigh the encoder in some cases. As a result, the multi-task dense prediction method may have more computation costs for the ViT-base backbone than the ViT-L backbone because of increased computation costs in the decoder. For instance, TaskExpert [3] have more computation cost for the ViT-base backbone (782GFLoPs) than the ViT-large backbone (622GFLoPS).
>
> [1] Yuqi Yang, Peng-Tao Jiang, Qibin Hou, Hao Zhang, Jinwei Chen, and Bo Li. Multi-task dense prediction via mixture of low-rank experts
>
> [2] Hanrong Ye and Dan Xu. Taskprompter: Spatial-channel multi-task prompting for dense scene understanding. In ICLR, 2022b.
>
> [3] Hanrong Ye and Dan Xu. Taskexpert: Dynamically assembling multi-task representations with memorial mixture-of-experts. In ICCV, 2023.

---

> > ### Comment · Reviewer_T3fK · 2024-11-24
> >
> > Thanks the authors for the response. My previous concerns have been addressed.
> >
> > I have two more questions to discuss with the authors.
> > 1. If I understand correctly, the inference of the proposed model depends on an initial Gaussian noise. So if different seeds are used in the experiments, the trained model would generate different predictions? Does this factor influence the overall performance?
> >
> > 2. Another question raised when I read the responses to other reviewers. In the response A4 to reviewer Cv9P, the authors claimed "Some of the previous methods, including PAD-Net, MTI-Net, and ATRC, are implemented based on HRNet in the original paper. For a fair comparison, we reimplement these methods based on ViT-Large, and the performance is also shown in Tab. 1." However, in the response to reviewer QTvQ, the authors mentioned " The performance of ATRC with ViT-large listed in Tab. 1 is reproduced in Taskprompter [2]." I also find that the additional results in revised Tab. 1 are the same as those reported in TaskPrompter. I suggest the authors give an explanation of how these numbers were obtained.

---

> > > ### Author Response · Authors · 2024-11-25
> > >
> > > Thank you for your comment. For your first question, the seed does
> > > influence the performance, though, the influence is ignorable. We list
> > > three results generated with the same trained model under the different
> > > initial seeds. For efficiency, we use the loss value to evaluate the
> > > performance of boundary detection.
> > >
> > >  | Method | Semseg $\uparrow$ | Parsing$\uparrow$ | Sal.$\uparrow$ | Nor. $\downarrow$ | Bound. loss $\downarrow$ |
> > >  | ------- | ------------- | ------------- | -------- | ------------ | ----------- |
> > >  | Seed 1 | 81.2058 | 69.6165  |  84.9360   |     13.5463  |      0.04273246 |
> > >  |  Seed 2   |   81.2060  |   69.6164 |   84.9360   |   13.5463 |   0.04273244 |
> > >  | Seed 3 | 81.2058  | 69.6165 | 84.9361   |   13.5463    |    0.04273246 |
> > >
> > > It can be seen that the difference in performance is less than 0.01%.
> > > This shows that our method is robust against different noise
> > > initialization.
> > >
> > > For the second question, we apologize for the unclarity we made. The
> > > correct statement should be: “**Some of the previous methods, including
> > > PAD-Net, MTI-Net, ATRC, MQTransformer, and DeMT, did not report their
> > > performances based on ViT-large backbone in their original papers. For a
> > > fair comparison, we borrow performances of PAD-Net, MTI-Net, and ATRC
> > > reimplemented with ViT-large from TaskPrompter. In addition, we
> > > reimplement MQTransformer and DeMT based on the ViT-large backbone by
> > > ourselves and report their performances in the revision.**” We have
> > > revised this statement in the comment. Besides, in the revision in
> > > Tab.1, the performances of PAD-Net, MTI-Net, and ATRC have been noted
> > > that they are borrowed from TaskPrompter, which is exactly what we mean.

---

> > > > ### Comment · Reviewer_T3fK · 2024-11-25
> > > >
> > > > Thanks the authors for the response.
> > > >
> > > > For my first question, it is good to know that the seed has minor influence on the evaluation results. For the second one, the corrected statement makes sense, and I suggest the authors to avoid unclear descriptions in future work.
> > > >
> > > > I decide to keep my original rating.

---

> > > > > ### Author Response · Authors · 2024-11-27
> > > > >
> > > > > Thank you for your efforts that have greatly helped us improve the quality of the paper. If there are further questions, we will be more than happy to discuss them.

---

### Official Review · Reviewer_Cv9P · 2024-11-02

**Soundness:** 3
**Presentation:** 2
**Contribution:** 2
**Rating:** 6
**Confidence:** 4

**Summary:**

This submission proposes a network named TaskDiffusion based on the recent diffusion work (DDP[1]) for multi-task dense prediction. The TaskDiffusion consists of three proposed modules. Pixel-level encoder encodes an image to feature which is utilized as multi-level conditions for diffusion decoder. Cross-task label encoder encodes the label of different tasks into a shared feature as a denoising input. Cross-task diffusion decoder is used for cross-task feature interaction and prediction decoding. The evaluations on two benchmarks demonstrate the effectiveness of the proposed method.

**Strengths:**

- This submission proposed a new diffusion model for MTL dense prediction, inspired by DDP.
- The empirical evaluations on two datasets (PASCAL-Context and NYUD-v2) suggest that the proposed method could improve the MTL performance with 4 and 5 tasks.

**Weaknesses:**

- The related works of MTL dense prediction are quite old. Some recent research[1-4] related to MTL dense prediction should be considered as references.
- The claim “This strategy utilizes the diffusion model to model the task relations explicitly in a coarse-to-fine process during denoising, …” in lines 79-80 should be clarified more deeply. To be specific, the illustration of $A_i$ in Fig.3 is hard to comprehend. What is the relationship between a pair of task?
- Some experimental details should be corrected and provided. For instance, the original PASCAL-Context does not provide the label for the surface normal task. And what is the resolution of input?
- Some comparisons are not fair. For example, the results of HRNet18 or HRNet48 should not be compared to ViT-large, since the backbone affects the performance remarkably.
- Some magnificent results are missing. In Table.1 and 2, the single task should be considered a baseline to figure out the impact of the negative transfer and the improvement.  And $\Delta_m$ should be provided in Tables 1 and 2.
- The presentation of Table.3 and 4 should be improved. How the $\Delta_m$ is calculated? If it is based on the single task baseline, it should be provided.

[1] Exploiting Diffusion Prior for Generalizable Dense Prediction, CVPR 2024

[2] UNITE: Multitask Learning With Sufficient Feature for Dense Prediction, IEEE Transactions on Systems, Man, and Cybernetics: Systems

[3] Rethinking of Feature Interaction for Multi-task Learning on Dense Prediction, arXiv 2312.13514

[4] MultiMAE: Multi-modal Multi-task Masked Autoencoders, ECCV 2022

**Questions:**

Please refer to weakness.

---

> ### Author Response · Authors · 2024-11-21
>
> **To reviewer Cv9P:**
>
> Thank you for your valuable comments. We will address them in detail in the following.
>
> **Q1**: The related works of MTL dense prediction are quite old. Some recent research[1-4] related to MTL dense prediction should be considered as references.
>
> **A1**: Thank you for your advice. We have added these works to the **Related Work** in the revision.
>
> **Q2**: The claim “This strategy utilizes the diffusion model to model the task relations explicitly in a coarse-to-fine process during denoising, …” in lines 79-80 should be clarified more deeply. To be specific, the illustration of $A_i$ in Fig.3 is hard to comprehend. What is the relationship between a pair of task?
>
> **A2**: Thanks for the suggestion. To utilize the diffusion model to model the task relation, we introduce the pixel-wise attention map $A_i^{s, p}$ generated in the diffusion process. As introduced in Sec.3.2, $A_i^{s, p}$ is a $H\times W$ map that represents the pixel-wise attention map from task $s$ to another task $p$. $A_i^{s, p}$ can fuse the task-specific feature from different tasks to the target task dynamically, and can act as an explicitly modeled task relations. During the diffusion process, $A_i^{s, p}$ will focus on a more specific area as shown in Fig. 3, which can support our claim in lines 79-80. In Fig.3, we visualize the $A_i^{s, p}$ mentioned in Sec.3.2 for specific $s$, $p$ and $i$. We annotate task pairs on the bottom of each figure. For example, in the visualized $A_i^{s, p}$ in the left-top figure in Fig. 3, $s$ represents the task index for the saliency detection and $p$ represents the index for boundary detection.
>
> **Q3**: Some experimental details should be corrected and provided. For instance, the original PASCAL-Context does not provide the label for the surface normal task. And what is the resolution of input?
>
> **A3**: Thank you for your suggestions. We follow the previous works [1, 2] and use the PASCAL-Context dataset with labels on five tasks. The original PASCAL-Context has annotations for semantic segmentation, human part segmentation, and edge detection. The surface normals and saliency labels are obtained from [3]. These labels are distilled from pre-trained state-of-the-art models. The input resolution is set to 512, which also follows the previous works. We have added this information in the Sec.4.1 of the revision.
>
> **Q4**: Some comparisons are not fair. For example, the results of HRNet18 or HRNet48 should not be compared to ViT-large, since the backbone affects the performance remarkably.
>
> **A4**: In Tab. 1, we compare the performance of our method and previous SoTA. Some of the previous methods, including PAD-Net, MTI-Net, ATRC, MQTransformer, and DeMT, did not report their performances based on the ViT-large backbone in their original papers. For a fair comparison, we borrow performances of PAD-Net, MTI-Net, and ATRC reimplemented with ViT-large from TaskPrompter. In addition, we reimplement MQTransformer and DeMT based on the ViT-large backbone by ourselves and report their performances in the revision. It can be seen that our method can still surpass the performance of these previous SoTA when they are reimplemented with ViT-Large. In addition, for a clearer and fairer comparison with the previous SoTA, we modify Tab.1 and Tab.2 in the revision and make sure all the compared methods are based on ViT-large.

---

> ### Author Response · Authors · 2024-11-21
>
> **Q5**: Some magnificent results are missing. In Table.1 and 2, the single task should be considered a baseline to figure out the impact of the negative transfer and the improvement. And should be provided in Tables 1 and 2.
>
> **A5**: For PASCAL-Context, we list below the STL performance and $\Delta_m$ in Tab. 7.
> |Method   | Semseg $\uparrow$ | Parsing$\uparrow$ | Sal.$\uparrow$ | Nor. $\downarrow$ | Bound.$\uparrow$ | $\Delta_m$ $\uparrow$
>  | --- | --- |  -- |  -- | --- | --- | - |
>  | ViT-L Single Task    |  81.62   |  72.21   |  84.34    | 13.59  |  76.79  |- |
>  | InvPT   |   79.03  | 67.61 | 84.81 | 14.15 | 73.00 | -3.61 |
>  | TaskPrompter | 80.89 | 68.89 | 84.83 | 13.72 | 73.50 | -2.03 |
>  | TaskExpert | 80.64 | 69.42 | 84.87 | 13.56 | 73.30 | -1.74 |
>  | ours  | 81.21  | 69.62  |  84.94 |  13.55 | 74.89 | **-1.11** |
>
> Also, the STL performance for NYUD-v2 is listed as follows.
>
>  | Method | Semseg $\uparrow$| Depth.$\downarrow$ | Nor. $\downarrow$ | Bound.$\uparrow$ |    $\Delta_m$ $\uparrow$
> | ----- | ----| ---| -- | - | --|
> |  ViT-L Single Task  |  56.77  | 0.5141 |  18.56 | 78.93  |   -
>  | InvPT | 53.56 | 0.5183 | 19.04 | 78.10 | -2.52 |
>  | TaskPrompter | 55.30 | 0.5152 | 18.47 | 78.20 | -0.81 |
>  | TaskExpert | 55.35 | 0.5157 | 18.54 | 78.40 | -0.84 |
>  | ours | 55.65 | 0.5020 | 18.43 | 78.64 | **0.18** |
>
> These results have been added to Tab.1 and Tab.2 in the revision.
>
> **Q6**: The presentation of Table.3 and 4 should be improved. How the $\Delta_m$ is calculated? If it is based on the single task baseline, it should be provided.
>
> **A6**: As proposed by the previous work [3], $\Delta_m$ is the average per-task performance drop of a task $m$ with respect to the single-task learning baseline $b$. The calculation of $\Delta_m$ can be formulated as follows:
> $\Delta_m=\frac{1}{T}\sum_{i=1}{T}(-1)^{l_i}(M_{m, i}-M_{b, i})/M_{b_i},$
> where $l_i=1$ if a lower value means better for measure $M_i$ of task $i$. The performance of the single-task learning baseline has been added to Tab.3 in the revision.
>
> [1]Hanrong Ye and Dan Xu. Taskprompter: Spatial-channel multi-task prompting for dense scene understanding. In ICLR, 2022b.
>
> [2]Simon Vandenhende, Stamatios Georgoulis, and Luc Van Gool. Mti-net: Multi-scale task interaction networks for multi-task learning. In ECCV, pages 527–543. Springer, 2020.
>
> [3]Kevis-Kokitsi Maninis, Ilija Radosavovic, and Iasonas Kokkinos. Attentive single-tasking of multiple tasks. In CVPR, pages 1851–1860, 2019

---

> ### Comment · Reviewer_Cv9P · 2024-11-22
>
> Thank you! Most of my concerns have been addressed. I still have some questions about the $A_i$ in Fig.3. For example, why does the saliency task focus on the human area of the boundary's feature? And why human parsing concentrate on the corners of the semseg's feature? For the other two attention maps, I still have the same questions. More explanations and clarifications would be better.

---

> > ### Author Response · Authors · 2024-11-23
> >
> > Thank you for your comment.
> > First of all, we apologize for a typo mistake in our revision in Fig.3.
> > The annotation for the right-bottom figure should be "Human parsing $\rightarrow$ Saliency".
> > We have corrected this annotation and modified the visualized content for clarity in the new revision.
> >
> > To generate predictions with high quality, the neural network needs to extract various types of features from the original image.
> > For instance, some features focus on boundaries, while others focus on the subject of the image.
> > By synthesizing these features from different aspects together, the neural network can make an accurate prediction in different tasks.
> > As a result, the cross-task attention map may not be directly correlated to the predictions, though, it will focus on the features that help improve the accuracy of the predictions.
> >
> > For example, in the left-top attention maps, saliency detection needs to detect the salient part of this image, which is the person.
> > In addition, the feature of boundary detection needs to distinguish the area inside the boundary from the outside.
> > It will be useful for saliency detection to leverage the features inside the boundary of the person to distinguish the salient part of the human body from the background.
> > As a result, the saliency detection task will pay more attention to the features of humans.
> >
> > In addition, in the left-bottom attention maps, it can be seen that human parsing leverages the features from the background in semantic segmentation.
> > Since the features of the person's area are required to be similar in different body parts, it is natural to ignore these features in human parsing, and the attention map also assigns low attention values to these areas.
> > When it comes to the high attention value at the corner, we first need to delve into the details of inference.
> > During inference, we pad the image to a resolution of 512$\times$512 and the attention map assigns higher values to the padded area, which is also the background of this image.
> > In the visualization, we cut the padded area for clarity, but some of the high-attention values remain in the cut image, which is the high-attention value at the corner.
> >
> > In the right-top attention maps, they focus on the shadow on the ground and the bench on the left, where the boundaries are annotated around these areas.
> > The semantic segmentation task can leverage these features inside the boundary extracted from the boundary detection branch to distinguish parts of the background from the area of the person and the dog.
> > At last, the right-bottom attention maps focus on the hand of the person, which is also one of the segmentation targets of human parsing.

---

> ### Comment · Reviewer_Cv9P · 2024-11-26
>
> Thank you for your response. I do recommend the authors add the corrected statements and analysis to the next revision. I will raise my score.

---

> > ### Author Response · Authors · 2024-11-27
> >
> > Thank you for your comment.
> > We have already added these analyses to Sec A.4 in the appendix in the revision.
> > In addition, the typo mistake has also been corrected in the revision.
> > Please feel free to discuss with us if you have further questions.

---

### Official Review · Reviewer_YCpJ · 2024-11-03

**Soundness:** 2
**Presentation:** 2
**Contribution:** 2
**Rating:** 6
**Confidence:** 4

**Summary:**

This paper introduces a diffusion-based model for multi-task dense prediction. During each training step, it first projects multi-task labels into a unified feature space and applies noise within that space. It then implements a cross-task diffusion decoder to iteratively predict the denoised features. Experimental results demonstrate performance improvements over previous methods.

**Strengths:**

- The study of multi-task learning is an important direction.

- We see performance improvement on PASCAL and NYUD compared to previous SOTA methods.

- The algorithm scheme is helpful for understanding.

- Extensive experiments have been conducted to thoroughly evaluate the model performance.

**Weaknesses:**

- The figures lack sufficient detail. The task interaction module and level interaction modules should be included in Figure 2 for better clarity.

- In Algorithm 2, the term "masks_pred" is not mentioned in the context. Should it be "m_preds"?

- The novelty is somewhat limited, as previous research has already demonstrated that diffusion-based multi-task learning is feasible.

**Questions:**

- Can you visualize the cross-task map at different steps to better understand it?

---

> ### Author Response · Authors · 2024-11-21
>
> **To reviewer YCpJ:**
>
> Thank you for your valuable comments. We will address your concerns in the following point-by-point.
>
> **Q1**: The figures lack sufficient detail. The task interaction module and level interaction modules should be included in Figure 2 for better clarity.
>
> **A1**: Thank you for your advice. As suggested by the reviewer, we add a more detailed model architecture of task-interaction modules and level-interaction modules in Fig. 4 in the revision, which is an extension of Fig. 2. We hope this can make the understanding of the proposed method clearer.
>
> **Q2**: In Algorithm 2, the term “masks\_pred” is not mentioned in the context. Should it be “m\_preds”?
>
> **A2**: Thanks for your comment. The “masks\_pred” in Algorithm 2 should be “m\_preds”. We have corrected this in the revision.
>
> **Q3**: The novelty is somewhat limited, as previous research has already demonstrated that diffusion-based multi-task learning is feasible.
>
> **A3**: Although diffusion-based multi-task learning has been proven to be feasible by previous methods, some challenges remain to be solved. These challenges include the lack of digging task relations, the inefficiency resulting from the iterative denoising process, and the cumbersome encoding design for labels of different tasks. To solve these challenges, we present a novel multi-task diffusion pipeline.
>
> Our novelty can be summarized in two points. Firstly, inspired by the outstanding ability of the diffusion models to capture the underlying distribution of prediction for each task, we want to leverage the diffusion models in the key problem of multi-task learning, which is the generation of cross-task relations. With the help of our proposed cross-task diffusion decoder, our model can model the cross-task relations explicitly in a coarse-to-fine manner, and results in better overall performance than the previous diffusion-based multi-task dense prediction method, DiffusionMTL, as shown in Tab.1 in the paper. In addition, by denoising all the tasks jointly, our proposed cross-task diffusion decoder can speed up the overall diffusion process in inference and needs less computation cost than DiffusionMTL as shown in Tab.1.
>
> Secondly, we propose the cross-task label encoder to unify the encoding of different tasks. Specifically, it encodes the continuous or discrete spatial map to a shared feature space. The proposed label encoder can get rid of the cumbersome task-specific encoding and encode the discriminative information from different tasks to improve the overall performance.
>
> **Q4**: Can you visualize the cross-task map at different steps to better understand it?
>
> **A4**: Thanks for the suggestion. We visualize the cross-task map at different steps in Fig.6 and analyze it in Sec.A.3 in the revision.

---

> > ### Comment · Reviewer_YCpJ · 2024-11-30
> > **Thank you for the response**
> >
> > I am happy with the response provided by the authors. I also appreciate the efforts to improve the paper based on my comments. I will keep my initial rating.

---

> > > ### Author Response · Authors · 2024-12-02
> > >
> > > Thank you for your comment and your efforts that have helped us improve the quality of the paper.
> > > If there are further questions, we will be very happy to discuss them.

---

### Author Response · Authors · 2024-11-21

We would like to thank all the reviewers for their constructive feedback on our paper.
Following the reviewers' advice, we revised our paper, and all the revised parts are marked as **blue**.

---

### Meta-Review · Area_Chair_BM3P · 2024-12-22

**Metareview:**

The paper proposed a novel framework that benefits from the diffusion model for multi-task dense prediction. The reviewers appreciated the contribution proposed in the method and the superior results achieved by comparing with some existing state-of-the-art performances.  AC agrees with the positive comments from the reviewers and recommends accepting the paper as a poster.

**Additional Comments On Reviewer Discussion:**

The reviewers requested clarification about the key contribution and additional experiments and analysis. Overall, the reviewers' rebuttal has convinced them.

---

### Decision · Program_Chairs · 2025-01-22

Accept (Poster)